# An analysis of tissue-specific alternative splicing at the protein level

**Jose Manuel Rodriguez**[1], **Fernando Pozo**[2], **Tomas di Domenico**[2], **Jesus Vazquez**[1,3], **Michael L. Tress**[2]*

**1** Cardiovascular Proteomics Laboratory, Centro Nacional de Investigaciones Cardiovasculares (CNIC), Calle Melchor Fernandez, Madrid, Spain, **2** Bioinformatics Unit, Spanish National Cancer Research Centre (CNIO), Calle Melchor Fernandez, Madrid, Spain, **3** Centro de Investigación Biomédica en Red de Enfermedades Cardiovasculares (CIBERCV), Madrid, Spain

* mtress@cnio.es

**Data Availability Statement:** All relevant data are within the manuscript and its Supporting Information files.

**Funding:** This study was supported by the National Institutes of Health (https://www.nih.gov) grant

## Abstract

The role of alternative splicing is one of the great unanswered questions in cellular biology. There is strong evidence for alternative splicing at the transcript level, and transcriptomics experiments show that many splice events are tissue specific. It has been suggested that alternative splicing evolved in order to remodel tissue-specific protein-protein networks. Here we investigated the evidence for tissue-specific splicing among splice isoforms detected in a large-scale proteomics analysis. Although the data supporting alternative splicing is limited at the protein level, clear patterns emerged among the small numbers of alternative splice events that we could detect in the proteomics data. More than a third of these splice events were tissue-specific and most were ancient: over 95% of splice events that were tissue-specific in both proteomics and RNAseq analyses evolved prior to the ancestors of lobe-finned fish, at least 400 million years ago. By way of contrast, three in four alternative exons in the human gene set arose in the primate lineage, so our results cannot be extrapolated to the whole genome. Tissue-specific alternative protein forms in the proteomics analysis were particularly abundant in nervous and muscle tissues and their genes had roles related to the cytoskeleton and either the structure of muscle fibres or cell-cell connections. Our results suggest that this conserved tissue-specific alternative splicing may have played a role in the development of the vertebrate brain and heart.

## Author summary

We manually curated a set of 255 splice events detected in a large-scale tissue-based proteomics experiment and found that more than a third had evidence of significant tissue-specific differences. Events that were significantly tissue-specific at the protein level were highly conserved; almost 75% evolved over 400 million years ago. The tissues in which we found most evidence for tissue-specific splicing were nervous tissues and cardiac tissues. Genes with tissue-specific events in these two tissues had functions related to important cellular structures in brain and heart tissues. These splice events may have been essential for the development of vertebrate heart and muscle. However, our data set may not be

number 2 U41 HG007234 (FP and MLT), the
Spanish Ministry of Science, Innovation and
Universities (https://www.ciencia.gob.es) grants
BIO2015-67580-P and PGC2018-097019-B-I00
(JMR and JV), the Carlos III Institute of Health-
Fondo de Investigación (https://www.isciii.es)
Sanitaria grant PRB3, IPT17/0019 - ISCIII-SGEFI /
ERDF, ProteoRed (JMR and JV), the Fundació
MaratóTV3 (https://www.ccma.cat) grant 122/C/
2015 (JMR and JV) and "la Caixa" Banking
Foundation (https://obrasociallacaixa.org) project
code HR17-00247 (JMR and JV). The funders had
no role in study design, data collection and
analysis, decision to publish, or preparation of the
manuscript.

**Competing interests:** The authors have declared
that no competing interests exist.

representative of alternative exons as a whole. We found that most tissue specific splicing
was strongly conserved, but just 5% of annotated alternative exons in the human gene set
are ancient. More than three quarters of alternative exons are primate-derived. Although
the analysis does not provide a definitive answer to the question of the functional role of
alternative splicing, our results do indicate that alternative splice variants may have played
a significant part in the evolution of brain and heart tissues in vertebrates.

## Introduction

Almost all multi-exon genes are able to undergo alternative splicing [1,2] via a range of mecha-
nisms which include exon skipping, alternative splice site usage and alternative promoter and
poly-A usage. This is reflected in the human reference set; at present human coding genes are
annotated with an average of four distinct gene products [3]. Recent studies suggest that
human coding genes generate on average more than ten alternative transcripts [4,5]. Assum-
ing that all, or almost all of these transcripts are translated into functional alternative splice iso-
forms, we would expect the overall protein population to increase 10-fold from 20,000 (the
number of human coding genes) to 200,000. This increase alone would have profound biologi-
cal consequences. However, most proteins do not work in isolation but interact with other
proteins, often as part of large complexes. If we take into account all the possible interactions
of these distinct proteins [6], we would be likely to see an exponential increase in the number
of cellular functions.

There has been much investigation at the transcript level to try to elucidate a role for alter-
native splicing and there is some indication that it may play a role in tissue differentiation.
Approximately two thirds of alternative splice events have been shown to have tissue-specific
differences. Wang *et al* [1] identified over 22,000 tissue-specific alternative transcript events
and showed that between 47 and 65% of alternative events were tissue-specific depending on
the type of splice event, while Gonzalez-Porta et al [7] found that the major transcript varied
according to conditions across more than 60% of coding genes.

However, it seems that not all tissue-specific splice patterns are conserved across species.
Merkin *et al* [2] found that despite the abundant evidence for tissue specificity of alternative
transcripts the patterns of tissue-specific alternative splicing were only conserved in a few tis-
sues between mammalian species and birds. Reyes et al. analysed tissue-specific splicing at the
transcript level across six primate species [8]. They found that only a small number of exons
had conserved splicing patterns. These exons with conserved patterns were enriched in
untranslated regions and the protein coding regions were enriched in disordered regions.
Meanwhile most tissue-specific alternative exons differed in their usage between species. They
postulated that the different usage of exons was behind the tissue-specific "rewiring" of pro-
tein-protein interaction networks postulated by many groups [9,10] that would be essential for
morphological differences between different species.

More recently, results from the large-scale GTEx consortium found that 84% of the vari-
ance between tissues was due to gene expression rather than alternative splicing [11] with the
strong suggestion that at least a certain proportion of tissue-specific alternative splicing is sto-
chastic [11]. A re-analysis of the GTEx data [12] found that 50% of genes had tissue-specific
transcripts, but that most tissue-dependent splicing events would not affect proteome com-
plexity of the cell since they involved untranslated exons.

Little research has been carried out into tissue-specific alternative splicing at the protein
level. Examples of protein level tissue specificity have been highlighted in analyses of individual

research papers [13,14], but there are no large-scale analyses of tissue specificity at the protein level. One reason for this is that proteomics experiments detect many fewer alternative isoforms than would be expected [15,16], It is not clear why it is so hard to detect alternative protein isoforms. Although most alternative transcripts seem to be processed by the ribosome [17], it has been shown that transcript level differences between species decrease post-translation [18]. Alternative isoforms that are not detected in proteomics experiments could be expressed in quantities too low for mass spectrometry detection, or in fewer tissues, they could have a shorter cellular half-life, or ribosome control mechanisms [19] could reduce their translation.

*In vitro* experiments have suggested that most alternative isoforms would lose or change more than 50% of their binding partners [6] relative to the main protein isoform [15,20]. Such gross changes in interaction partners suggest that most alternative isoforms would be more than just minor variants of the main isoforms. However, it is difficult to know to what extent such *in vitro* experiments are representative of the cellular proteome.

The large-scale proteomics study of 30 human tissues and hematopoietic cells carried out by Kim et al [21] remains the best source of tissue level proteomics data, in part because it was carried out with replicates. The data from the Kim experiments has been analysed on a number of occasions [22–24], however no study has investigated tissue-specific splicing of alternative isoforms in detail. The original study highlighted distinct isoforms of *FYN* protein tyrosine kinase in brain and haematopoietic cells [21], while Wright *et al* suggested that most tissue-specific alternative splicing was in testis without revealing details [22]. The other two studies detailed evidence for tissue-specific alternative splicing in just a few genes mostly localised to brain and heart tissues [23] or to heart and testis [24].

Here we carried out an *in-depth* study of tissue-specific alternative splicing in tissues from the Kim *et al* proteomics experiments and contrasted it with data from a large-scale transcriptomics analysis. We find that there is strong evidence for tissue-specific splicing at the protein level in a minority of genes, and that these tissue-specific protein isoforms are generally found in muscle or nervous tissues. Almost three quarters of tissue-specific splice events detected at the protein level are conserved all the way back to jawed vertebrates.

## Results

### Proteomics evidence for alternative splicing

We first used the peptides identified in the proteomics experiment to define the set of alternative splice events that would be used in the analysis. For this set we required that each side of a splice event be supported by a minimum of three PEDs (peptides from different experiments, see Materials and Methods section). Since each side of a splicing event is different, we defined the two sides as either "main" and "alternative". The main side of each splicing event is the side supported by most PEDs (in the case of the proteomics experiments), or most RNAseq reads (in the case of the transcriptomics experiments). From the peptide data we were able to define a set of 255 alternative splicing events that came from 217 genes (see **S1 Table**). This dataset (ASE255) was used for all subsequent analyses.

### Evidence for tissue-specific splicing in proteomics experiments

Initially we searched for evidence of both tissue-specific and group-specific differences at the protein level for the ASE255 set. For each tissue or tissue group we compared the distribution of PEDs for the main and alternative sides of each splice event, and also between the events and the rest of the protein. When we compared expression at the tissue level only, 51 events had significant differences in expression in at least one of the 30 tissues. When tissues were

grouped, we found significantly different levels of expression for 87 splice events. In total 95 of the 255 events had evidence for either tissue or group-specific splicing (37.3%), while there were 43 events with significant differences at both tissue and group level.

At the level of individual tissues, frontal cortex (17 tissue-specific events, **S1 Fig**) had the highest evidence of significant tissue-specific splicing. Fetal and adult heart tissues also had more than ten splice events with evidence for significant tissue-specific splicing at the protein level. When tissues were combined into groups, nervous tissues (frontal cortex, fetal brain, spinal cord and retina) had the highest level of tissue-specific alternative splicing (**Fig 1**). In fact, more than 50% of the 87 events that had significant differences in group-specific splicing were group-specific in nervous tissues (**Fig 1**).

## Tissue specificity by event type

We classified the 255 splice events in the ASE255 set according to two different criteria, the mechanism of the splicing process and the effect the splice event has on the resulting proteins. We divided the splicing mechanisms into six types, skipped exons, mutually exclusively spliced exons, alternative 5′ splice sites, alternative 3' splice sites, alternative promoters and alternative poly-A. Skipped exons were most numerous (93 events) with alternative 3' splice sites a distant second (37 events).

We divided the effect the splicing has on the protein into seven groups. Firstly, deletions and insertions for which we had peptides for each side of the event were pooled as "Indels"

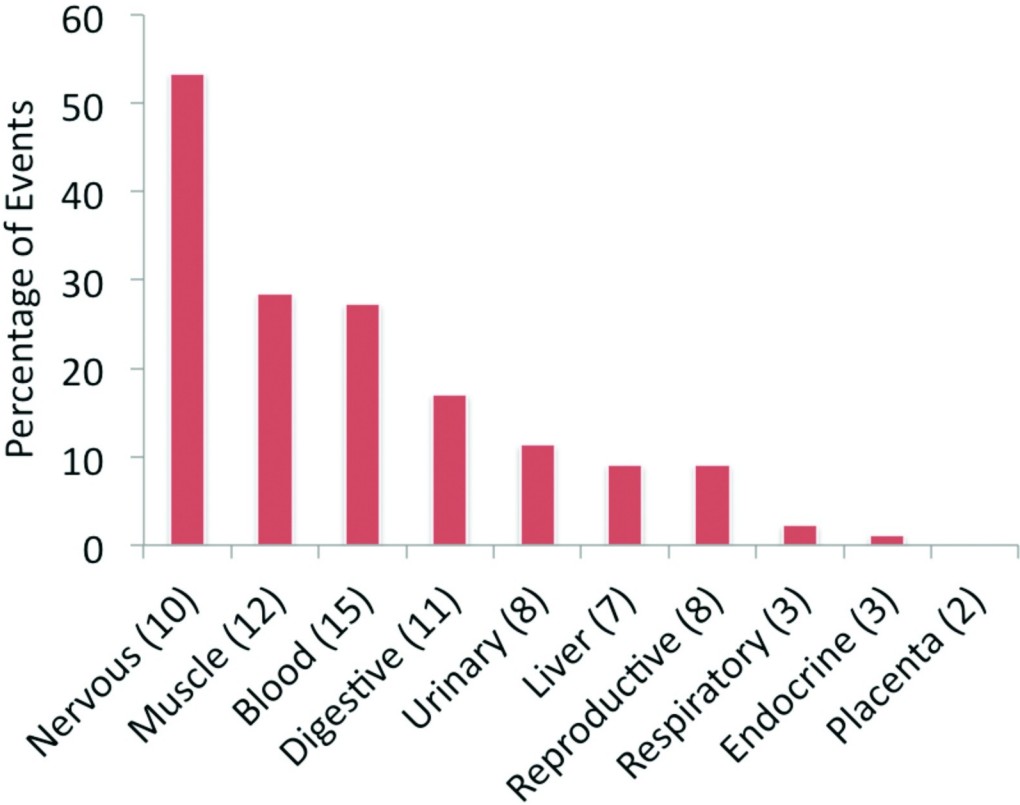

**Fig 1. Tissue-specific alternative splicing events at the proteomics level.** The percentage of significant tissue-specific alternative splicing events across the 10 proteomics tissue groups. The number of experiments for each tissue group is shown in the x-axis labels.

since we did not know which was the alternative isoform. Meanwhile smaller indels, those that were smaller than four amino acid residues, were pooled as "Micro-Indels". Some of these mini-indels were produced by micro-exons, but most were generated by NAGNAG splicing [25]. If the substitutions were homologous [26], we pooled them in a different "Homologous Substitution" category because we have found that these exons often behave differently to most substitutions [16,23]. The remaining substitutions were tagged by their position in the protein sequence (C-terminal, N-terminal or internal), while protein isoforms that did not share any amino acid sequences were pooled into a seventh group ("Two Proteins"). A total of 104 events were classified as Indels. Almost three quarters of the indels were generated from skipped exons. There were just 5 events in the "Internal Substitution" category and 4 in the "Two Protein" category.

We calculated the proportion of each type of event among events with evidence of tissue specificity from the proteomics experiments and also among events that did not have evidence of protein-level tissue specificity. Two types of splicing mechanisms generated considerably more tissue specific events than non-tissue specific events (mutually exclusive exon and alternative poly A), while two mechanisms (alternative 5´splice sites and alternative 3' splice sites) produced a substantially lower proportion of tissue specific isoforms (Fig 2A).

When classified by the effect at the protein level, tissue-specific and non-specific events were more or less proportionally distributed within each class (Fig 2B), but two types of events had distributions that were significantly different from the others. Homologous substitution events were enriched among tissue-specific events. Homologous exon substitutions made up a third of all events with tissue-specific differences, with a Fisher exact test of 0.0062 (against indels). By way of contrast Micro-indel category splice events had significantly fewer non-tissue specific events (Fisher exact test of 0.0018 against indels). More than half of the events in the Micro-indel category were formed via alternative 3' splice sites.

Half of the homologous swaps were produced from mutually exclusive exons, while the other half were produced from alternative Poly A and alternative promoters. The proportion of tissue specific events among non-homologous alternative Poly A events (all of which are C-terminal substitutions) and non-homologous alternative promoter events (all N-terminal substitutions) decreases considerably once the homologous substitution events have been removed. This suggests that sequence homology was more important than the mechanism of action of the splicing process in the gain of tissue specific splicing.

## Disorder and tissue specificity

Protein disorder has been strongly linked to alternative splicing [27] and to tissue-specific splicing in general [9]. We analysed the proportion of events with disorder in the ASE255 set and found that alternative exons in the set of splice events were enriched in disorder. A total of 43.6% of alternative exons were predicted to be disordered against 32.8% of the genes that made up the ASE255 set. However, there was no indication that disorder was related to tissue specificity either at the protein level, where 37.7% of tissue specific alternative regions were disordered against 48.1% of non-tissue specific regions, or at the transcript level (S2 Fig). Tissue specific skipped (cassette) exons are also depleted in predicted disordered regions in this set (S2 Fig).

## Most tissue-specific splicing occurs in ancient splice events

We manually curated the relative age of each of the 255 splicing events based on cross species evidence. Manually curated event ages were defined as primate-derived (up to approximately 75 million years old), as evolved during the eutheria/theria clades (between 75 and 160 million years ago), as evolved during the tetrapoda clade (evolved after sarcopterygii, between 160 and

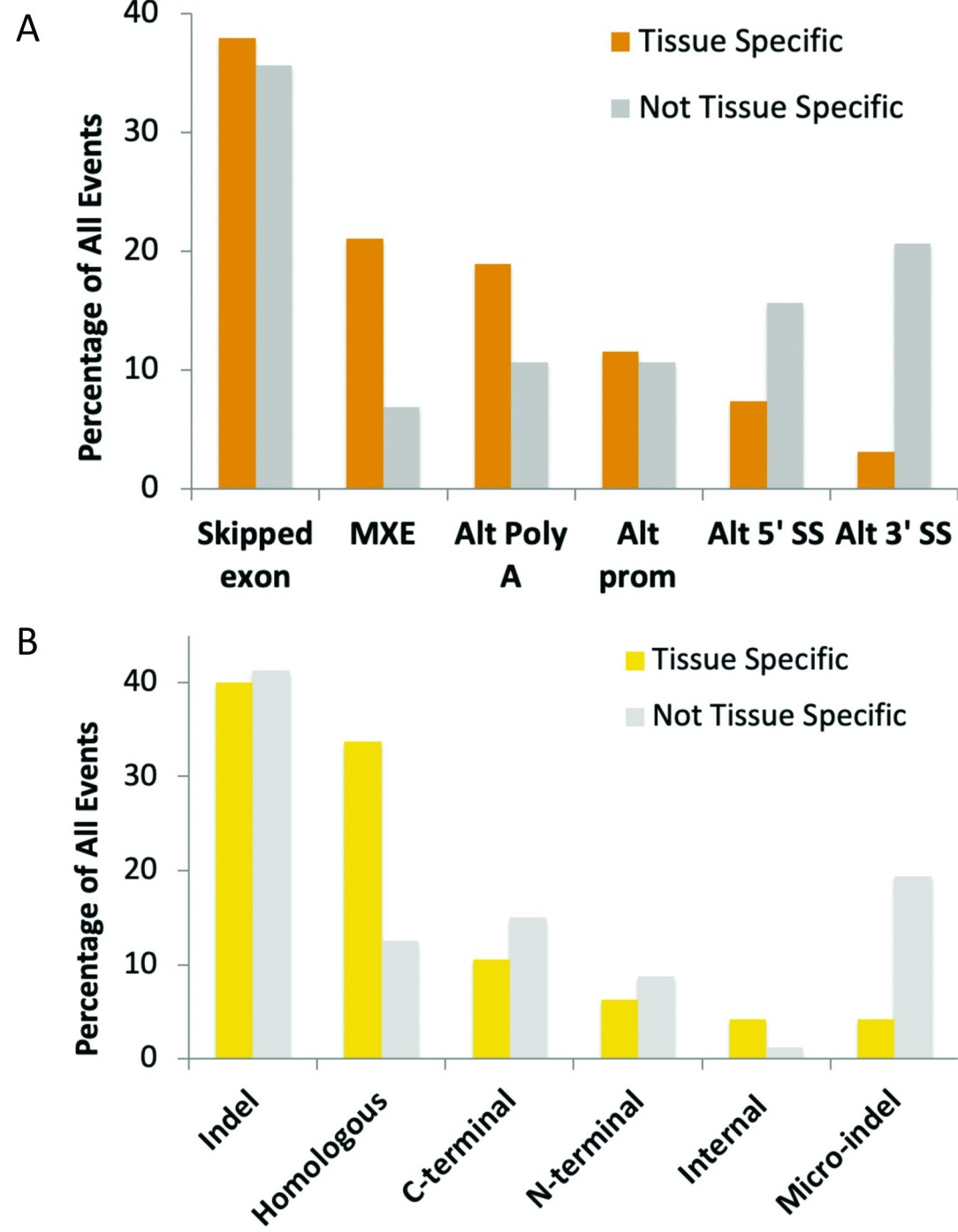

**Fig 2. Tissue-specific expression at the proteomics level by type of splice event.** (A) The breakdown of events by alternative splicing mechanism and tissue specificity, or lack of. The proportion of the 95 events that are tissue or group specific events are in orange. The proportions of event types among the 160 non-tissue specific events are shown in grey. (B) The relative proportions of tissue specificity of different protein level events. The proportion of the 95 tissue or group specific events that make up each event type is shown in yellow. The proportions of event types among the 160 non-tissue specific events are shown in light blue. All event types are defined in the main text.

400 million years ago) and as ancient (evolved before the sarcopterygii clade, more than 400 million years ago).

In order to compare the results against the whole genome we also estimated the relative age of alternative exons. Alternative exons were defined from their annotations in the APPRIS database and we analysed just those exons that had a minimum of 42 bases (see Materials and Methods section). This automatic estimation of alternative exon age is an approximation, but it does provide an idea of the relative proportions of the four age groups among alternative exons in the genome. We found that 76% of alternative exons in the human genome appeared in the primate clade, within the last 90 million years, while just 5.7% were more than 400 million years old (Fig 3).

By way of contrast to annotated alternative exons, alternative splice events detected in proteomics experiments were considerably more conserved: more than half of the alternative events in the ASE255 set evolved more than 400 million years ago and only 7.8% of the alternative events in the ASE255 set derived from the primate clade. We have previously shown that proteins from ancient gene families are more likely to be detected in proteomics experiments [28] and that there is little reliable proteomics evidence for primate-derived coding genes [28,29]. Hence, it is not surprising that we also found most evidence for ancient splice events and little evidence of alternative splicing events derived from the primate clade.

Not only was the set of alternative events detected at the protein level enriched in ancient events, but tissue-specific splice events were even more conserved. Almost three quarters

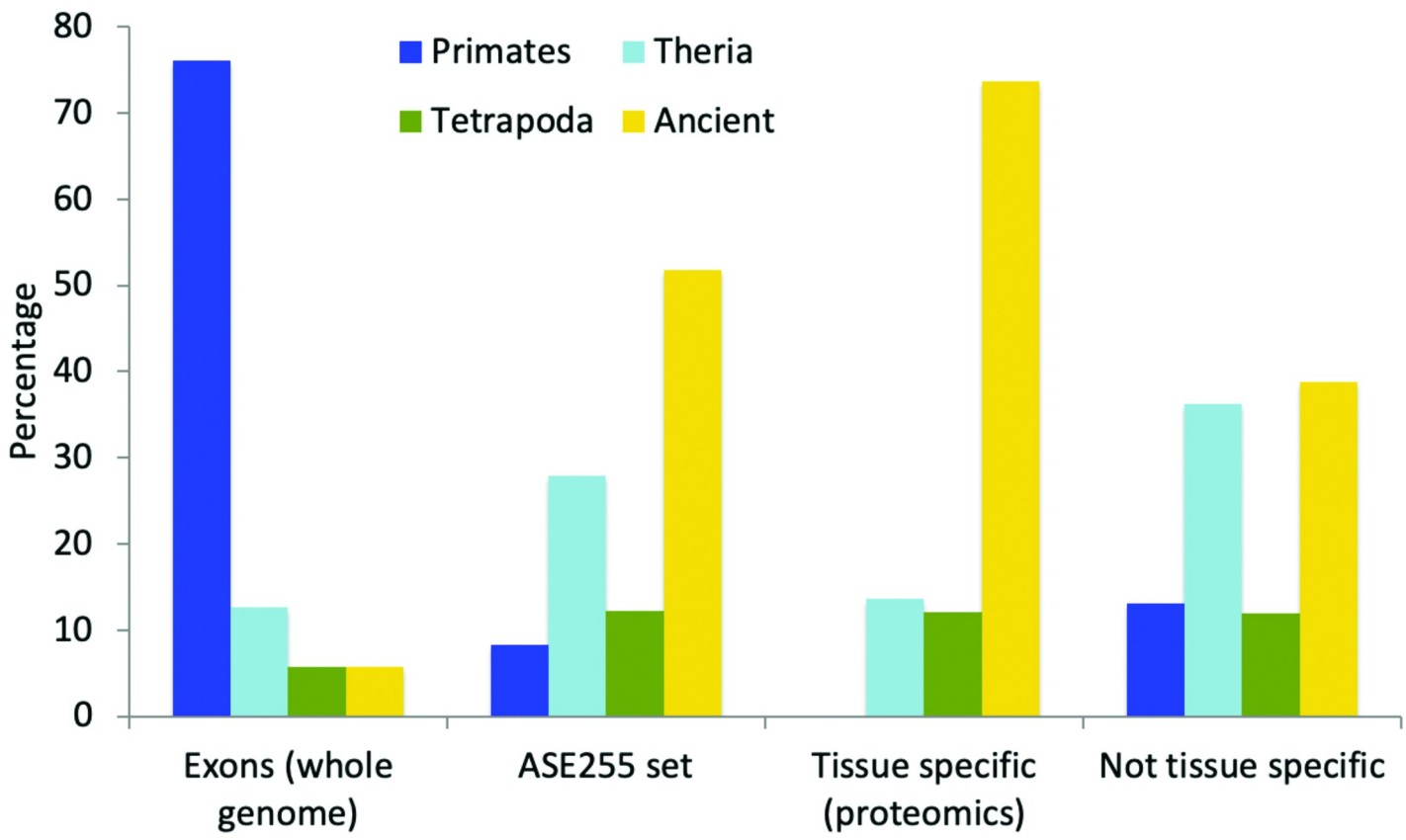

**Fig 3. The age of alternative exons versus subsets of splicing events detected in proteomics experiments.** "Exons (whole genome)" are all alternative exons in the human genome, "ASE255 set" is the set of 255 alternative splicing events detected in the proteomics analysis, "Tissue-specific (proteomics)" are the 95 events that have significant tissue or group-specific differences at the protein level and "Not tissue specific" are the 161 events that do not have tissue-specific enrichment in proteomics experiments.

(73.7%) of events with evidence of tissue specificity at the proteomics level evolved more than 400 million years ago (**Fig 4**). At the same time there is no evidence at all for tissue-specific splicing of primate-derived splice events at the protein level (**Fig 4**). Tissue-specific alternative splicing events detectable at the proteomics level are highly conserved.

### Functional relevance of tissue-specific splicing

We have previously found that alternative splice variants detected in proteomics experiments are enriched in functions related to the cytoskeleton [**16**]. We find similar results with the alternative splice events used in this analysis. This is not entirely surprising since proteomics analyses tend to be enriched in the most abundant proteins, including ribosome proteins and actin cytoskeleton-related proteins, and depleted in integral membrane proteins [**30**].

The top ten highest scoring GO terms for the 217 genes in the ASE255 set as a whole were all cytoskeleton-related and included *cell-cell adherens junction* (Benjamini-Hochberg adjusted q-value of 2.7E-08), *Z-disc* (6.8E-10), *structural constituent of muscle* (5.4E-10), *focal adhesion* (1.7E-07), and actin filament binding (1.5E-07). See **S2 Table** for more details. A total of 111 of the 217 ASE255 genes were labelled with terms related to the cytoskeleton. There was a strong relation between cytoskeleton-related genes, tissue specificity and conservation. Cytoskeleton genes with events that were tissue-specific at the proteomics level had a much higher proportion (82.1%) of events that evolved before or during the vertebrate clade (see **S3 Fig**). In fact, events in cytoskeleton genes were significantly more likely to be tissue specific (Fisher exact test, 0.0004) than non-cytoskeleton genes.

We also analysed the two subsets of events that had the most evidence of group-specific splicing at the protein level (**Fig 1**): those events with group-specific splicing at the protein level in nervous tissues (43 events from 37 genes) and those events with group-specific splicing at the protein level in muscle tissues (24 events, 18 genes). While the results were similar to the results for the ASE255 set, there were specific differences.

Terms for the nervous tissues specific events were related to adhesion, morphology and cellular communications and included *cadherin binding involved in cell-cell adhesion*" (0.002), *cell-cell adherens junction* (0.004), *stress fiber* (1.1E-04) and *plasma membrane* (4E-04), whereas terms for the events specific to muscle tissues were enriched in those terms more related to muscle, such as *Z-disc* (3.2E-05), *structural constituent of muscle* (2.3E-09), *actin filament organization* (3.2E-04) and *muscle thin filament tropomyosin* (5.7E-04). There was one term in common in the top 10 significantly enriched GO terms, *actin filament binding*. Tissue-specific splicing in the two sets of genes seemed to be related to cytoskeleton organization of the specific tissues. See **S2 Table** for more details.

### Comparison to tissue-specific splicing at the transcript level

We were able to map sufficient RNAseq reads to both sides of the splicing event for 248 of the 255 events. According to the criteria we used in our analysis (a difference of at least 1 standard deviation in expression levels between alternative events) there was evidence for group-specific differences in expression for a total of 159 of the 248 events (62.9%). This concurs with what has already been found by numerous groups; approximately two thirds of alternative splicing events are strongly tissue-specific at the transcript level [**1**].

The tissue group with the most evidence for group-specific expression for the events in the ASE255 set was digestive tissues (see **S4 Fig**), but nervous tissues, reproductive tissues, fat and muscle tissues also had high levels of group-specific expression at the transcript level. There was little evidence of tissue specificity at the transcript level among the 248 transcript level events we analysed in either liver or endocrine tissues.

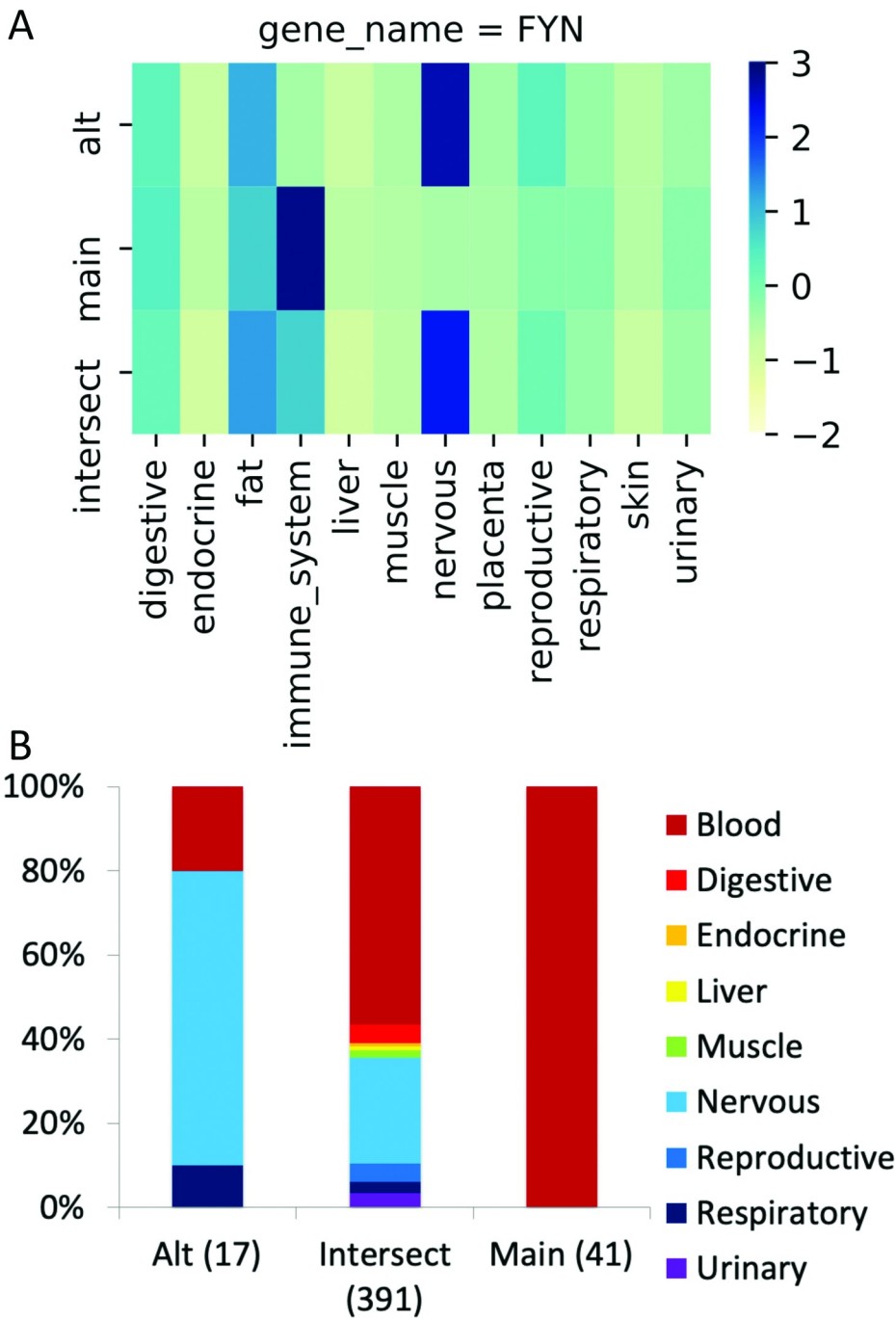

**Fig 4. The group-specific splicing event in *FYN*.** (A) Group-specific distribution of reads that support each side of the *FYN* splice junction (main, alternative) and those that support the common protein sequence (intersect) coloured by standard deviation from the mean; the darker the colour, the greater the positive standard deviation. The main transcript has more than one standard deviation of reads than the alternative for immune system tissues and the alternative transcript has more than one standard deviation of reads than the main transcript in nervous tissues. (B) The distribution of the PEDs from the proteomics experiments for the alternative and main isoforms ("Alt" and "Main") and those that map to the remainder of the common amino acid sequence ("Intersect"). The numbers of PEDs that belong to each group are shown in brackets. Fisher tests show that the alternative side of the event is significantly enriched in peptides from nervous tissues, just as in the RNAseq experiment. The main side of the event is significantly enriched in peptides from blood cells. Although both sides of the event are enriched in different tissue groups, *FYN* does not count twice towards the total of 99 cases in the PGE99 set because the main side of the event is enriched in blood cells.

## Coincidence of tissue-specific splicing at protein and transcript level

In order to make a direct comparison between the proteomics and transcriptomics data set, we had to generate a set of paired events from the splice events in the ASE255 set. For the comparison we required that the event was enriched in any of the tissue groups apart from blood. The tissue groups that we included had to be present in both proteomics and RNAseq analysis and the hematopoietic cells analysed in the Kim *et al* experiments [21] did not have a comparable tissue in the Uhlen *et al* analysis [31].

For the comparison we only considered those events in which one side of the event (main or alternative) was significantly enriched at the protein level in at least one of the grouped tissues. We left out events that were only significant at the tissue level and events in which the peptide evidence for the event was depleted rather than enriched. If an event was significantly enriched in more than one tissue group, we counted each tissue in which it was enriched as a distinct case. In total there were 99 cases of proteomics group-specific enrichment in which either the main or alternative side of a splice event was significantly enriched in a tissue group. An example is shown in **Fig 4**. The 99 cases came from 76 distinct events and are referred to here as the PGE99 set.

Reassuringly, we found that two thirds (66) of the protein level enrichments were also enriched in the same tissue group at the transcript level. The proportion of events significantly enriched at both the protein and transcript level differed substantially between tissue groups (**Fig 5A**). Many of the events enriched in muscle and nervous tissues at the transcript level were also enriched at the protein level; 32 of 78 splice events enriched at the transcript level in nervous tissues and 21 of 48 events enriched in muscle tissues were significantly enriched at the protein level. However, the same was not true of the other tissue groups. Only 7 of the 85 events enriched in digestive tissues and just 3 of 71 events enriched in reproductive tissues in the transcriptomics experiments were significantly enriched in the same tissues in proteomics experiments. There was significant enrichment for one event in both protein and transcript analyses in liver (*ACOX1*), one significantly enriched in respiratory tissues (*NEBL*) and one significantly enriched in urinary tissues (*TPM4*).

The higher proportions of events enriched at both transcript and protein-level in muscle and nervous tissues were statistically significant. Fisher's exact tests showed significant differences between nervous and placenta (p-value = 0.0005), nervous and digestive (p-value < 0.00001), nervous and reproductive (p-value < 0.00001), and even nervous and respiratory (p-value = 0.0495) and nervous and urinary tissues (p-value = 0.0272). The comparisons between muscle tissues and digestive, placenta, reproductive, respiratory and urinary tissues were similarly statistically significant.

In all but three of the 66 cases of significant enrichment in the same tissue at both the protein and transcript level the event could be traced back to a common ancestor with fish. Tissue specificity at the transcript level also seemed to be associated with conservation of splice events (**Fig 5B**). Events that were tissue-specific in the transcriptomics analysis were older than events without significant tissue-specific splicing (**Fig 5B**).The proportion of ancient (> 400 million years old) splice events that were tissue enriched at the transcript level was also significantly greater than the proportion of ancient splice events that were not tissue specific (Fisher exact test < 0.00001).

## Event age and tissue groups

To analyse why there was considerably more coincidence between protein level and transcript level tissue specific splicing in nervous and muscle tissues than in reproductive and digestive tissues we defined the sides of each significantly enriched transcript level event as either enriched (the side of the event with significantly more transcript evidence) or depleted.

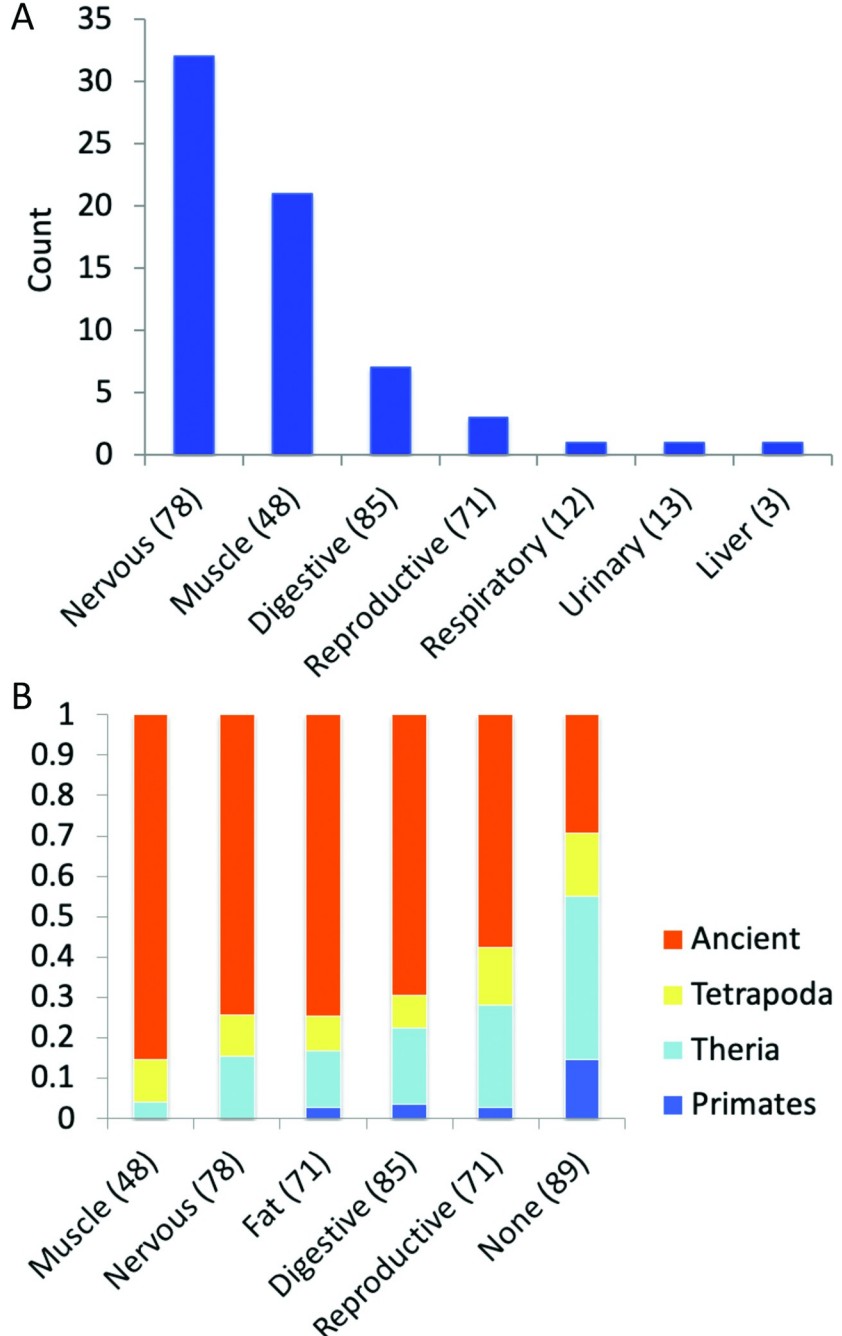

**Fig 5. Comparison of RNA-level tissue-specific events.** (A) The number of transcript-level enriched events also enriched at the protein level. Each bar shows the number of transcript-level tissue group enriched events that are enriched in the same tissue group in proteomics experiments. Enriched events were compared over the 9 tissue groups that coincided in both transcriptomics and proteomics experiments. The number of events that were tissue-specific in the transcriptomics experiments for each group is shown in brackets. (B) The age of the events enriched in RNAseq studies in the five most populated tissue groups and those not enriched at all (None). At the RNAseq level more of the muscle and nervous tissue enriched events are ancient than those in any other tissue. Results shown for tissues with a minimum of 48 tissue-specific enriched events.

With the two sides of each event defined we were able to sum the PEDs that supported each side of digestive, muscle, nervous and reproductive tissue specific events (S5 Fig). We found that there were significantly more PEDs for the transcript-enriched side of events than the depleted side in all four tissues (Fisher's exact tests: digestive 0.00001, muscle 0.0, nervous 0.0, reproductive 0.0007),

We calculated the percentages of supporting PEDs for the enriched and depleted sides of each individual event and generated scatter plots for each of the tissues (Fig 6). Most events had proportionally more supporting PEDs for the enriched side of the splice event than the depleted side in all four tissue groups. Where muscle and nervous tissues differed was that the PEDs that support the enriched side of the event were often highly enriched. Many of the points in muscle and nervous tissues fall a long way from the diagonal representing equal proportions of supporting PEDs (Fig 6) and the enriched side is supported by 100% of the PEDs in many events that are enriched in these tissues. By contrast none of the enriched sides of

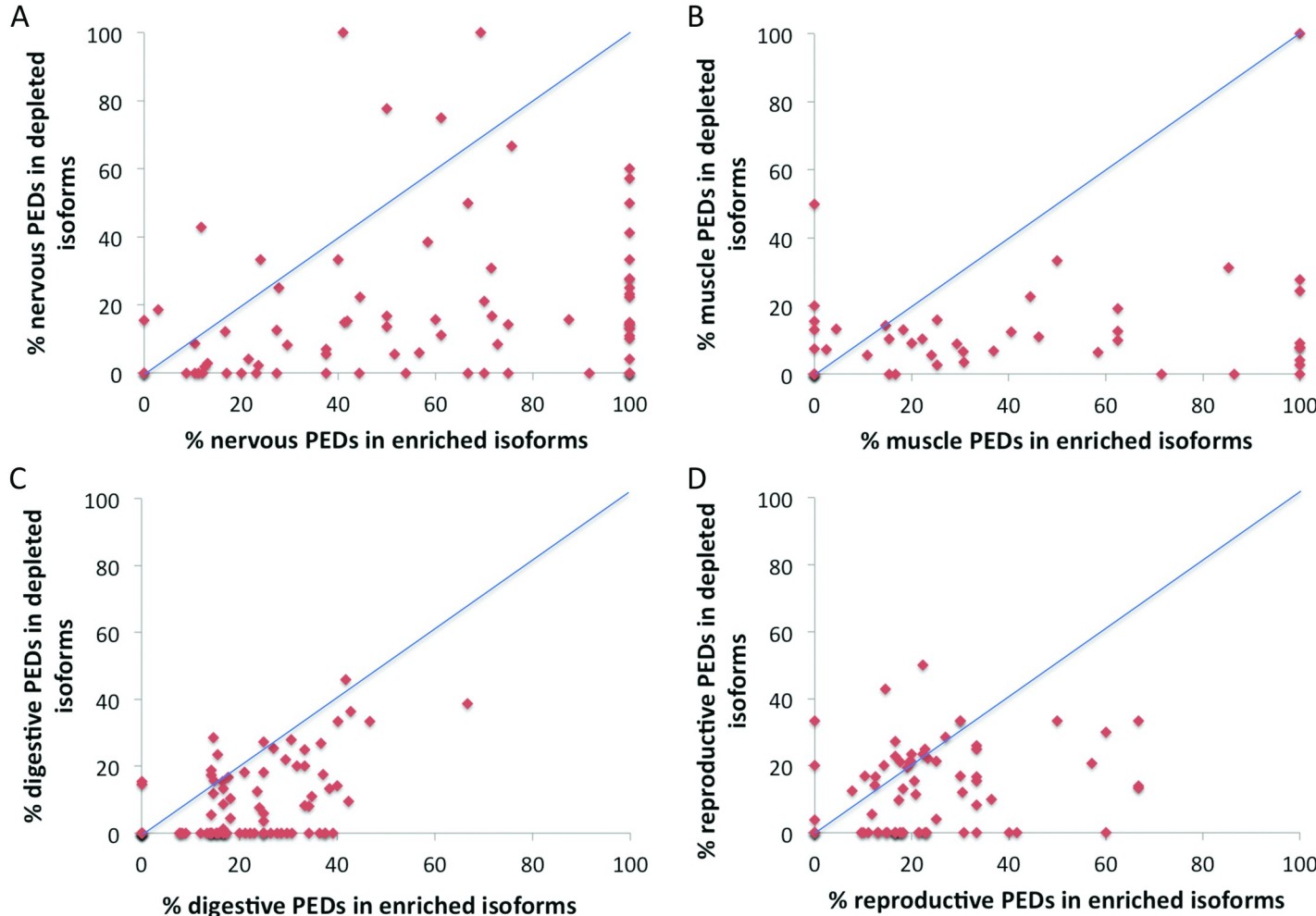

**Fig 6. Scatter plot of the percentages of PEDs supporting transcript level enrichment.** The figure shows scatter plots of the percentage of PEDs that support the selected tissue for the side of the event that is enriched in that tissue group in the transcriptomics experiments (X-axis) versus the percentage of PEDs that support the selected tissue for the side of the event depleted in the transcriptomics experiments (Y-axis). The diagonal line shows where the percentage of PEDs that support the side of the event that is enriched in the transcriptomics experiments is identical to the percentage of PEDs that support the side of the event that is depleted in that tissue group in the transcriptomics experiments. Events that have proportionally more PEDs on the transcript-enriched side of the event (those that agree with the transcriptomics evidence) ought to be below the line. Tissues shown are A. Nervous B. Muscle C. Digestive and D. Reproductive.

events in reproductive or digestive tissues is supported by more than 70% of total PEDs. The proteomics evidence suggests that many transcript-enriched events in digestive and reproductive tissues may also be enriched at the proteomics level, but the enrichment is often minimal, as seen by the clustering around the diagonal in these two tissues (**Fig 6**).

The data also suggests there may be a higher proportion of noisy splicing at the transcript level in reproductive tissues than in the other tested tissues, although it is difficult to draw firm conclusions from tissues which include testis. In more than a third (35.6%) of events that are reproductive-enriched at the transcript level there is as much or more evidence for the depleted side of the event at the protein level as there is for the enriched side. The 31 events that were tissue-specific at the transcript level in reproductive tissues that evolved most recently (since the split with fish) are not as a whole significantly enriched at the protein level (**S6 Fig**).

### Is there correlation between proteomics and transcriptomic data at the event level?

Since we had already calculated the percentage of PEDs that supported both sides of tissue specific splice events, we also calculated the percentage of RNAseq reads that supported splice events that were tissue specific in digestive, muscle, nervous and reproductive tissues. We determined the correlation between the percentage of PEDs and the percentage of RNAseq reads that supported each side of a splice event. Here there were also substantial differences between digestive, muscle, nervous and reproductive tissues here (**Fig 7**). The correlation between supporting PEDs and supporting reads was highest in nervous (0.799) and muscle (0.748) tissues and lowest in reproductive tissues (0.413). Plots of supporting reads against supporting PEDs are available for the four tissues (**S5 Fig**).

We used the whole of the ASE255 set to determine the correlation for each tissue by age of splice event (**Fig 7**). When comparing supporting PEDs and reads over all splice events the correlation will in part be due to gene expression rather than alternative splicing because we are including events that are not significantly tissue specific. This explains much of the high correlation among theria-derived events in reproductive tissues, for example. Despite this, it is clear that the correlation between proteomics and transcriptomics support is considerably worse for those splice events that arose in the primate clade. Correlation coefficients for primate-derived events (which make up more than three quarters of annotated alternative exons in the human gene set) ranged from 0.003 (in muscle) to 0.319 (digestive tissues).

### Caveats

There are a number of caveats to the comparison of proteomics and tissue level alternative splicing. Firstly, the analysis was carried out on a small number of alternative splicing events. This was inevitable because even large-scale mass spectrometry-based proteomics experiments detect few alternative isoforms reliably [**15**]. Secondly, even though we analysed those splice events with the most proteomics support, the relatively low numbers of discriminating peptides for each event limits statistical power and makes it harder to identify tissue specificity. A deeper exploration at the protein level is likely to show that there are tissue-specific differences for some events that we are not detecting.

Thirdly, the comparison between the proteomics and transcriptomics experiments was handicapped by the fact that the experiments were not paired (experiments did not come from the same individuals). Finally, we were only able to interrogate 9 tissue groups, and it is likely that other groups may also display further tissue-specific alternative splicing. For example, there was also substantial evidence for tissue-specific alternative splicing at the protein level in

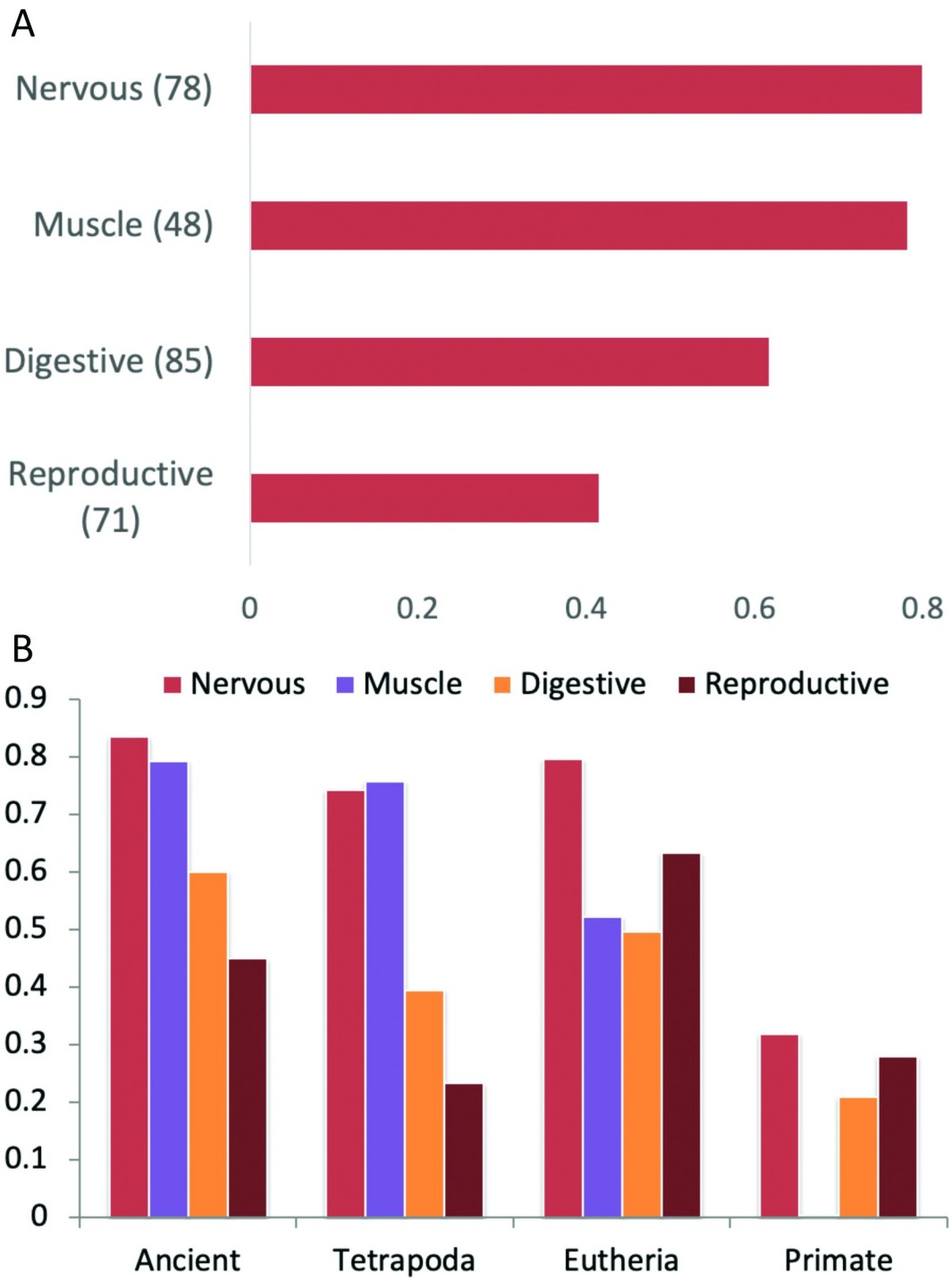

**Fig 7. Correlation between PED and RNAseq read support.** (A) Correlation between percentage PED and read support for those splice events enriched in grouped digestive, muscle, nervous and reproductive tissues at the transcript level. (B) Correlation between percentage PED and read support for splice events grouped by event age.

blood and at the transcript level in fat. Within these 9 tissue groups it was also harder to detect tissue-specific splicing at the protein level in tissue groups with fewer replicate experiments (placenta, endocrine and lung tissues).

Given the nature of proteomics experiments and the minimum requirement for three PEDs, we cannot demonstrate that any splice event is *not* tissue or tissue group specific, or even cell type specific. The lack of coverage and/or the different make up of tissues does play a role in the differences between the results at the transcript and protein level. For example, events in *ABI2*, and *ATP1B4* were enriched in brain tissue in the transcriptomics analysis and we identified multiple discriminating PSM for isoforms of in frontal cortex, but in both cases the differences were not significant at the protein level due to the lack of peptide coverage. Events in six genes, *FMN1*, *RAP1GAP*, *NECTIN1*, *IDE3B*, *FRMD5* and *ATP2B3*, that had apparent tissue specific differences at the protein level (two had PEDs in frontal cortex, one was apparently enriched in retina, one in fetal heart, one in adult heart, and one in pancreas) were left out of the analysis because one side of each event only had 2 PEDs. All eight of these events could be traced back to a common ancestor with fish.

## Conclusions

Transcript level studies consistently show that the majority of alternatively spliced exons are tissue specific. In this analysis we also find substantial tissue-specific alternative splicing also exists at the protein level. Just over a third of the 255 splice events validated in our proteomics analysis are significantly tissue specific.

Manual curation of the protein level tissue-specific splice events detected in our analysis found that almost three quarters had homologues in fish. No tissue specific splice event was primate-derived. This is in sharp contrast to the alternative exons in the human gene set, more than three quarters of which arose in the primate clade. Reyes *et al* found similar differences in tissue specific splice patterns: while a minority of conserved exons had large amplitude tissue-specific differences, exons with little variations in tissue specific usage were not conserved between species [8].

The stark differences in conservation between tissue specific splice events with evidence at the protein level and alternative exons in the human gene set mean that our results cannot be extrapolated to the whole genome. The lack of detectable tissue-specific splicing among recently evolved splice events suggests that primate-derived splice events are likely to have different tissue-specific behaviour and many may have low amplitude tissue differences, if they have any at all. The weak link between protein level tissue-specificity and recent splice events suggests that tissue-specific alternative splicing is unlikely to generate important species-specific differences.

The theory that alternative splicing might be responsible for large-scale tissue-specific protein-protein interaction networks [9,32] is based in part on evidence for tissue specific splicing, and in part on evidence that alternative exons are enriched in predicted disorder. While we find that alternative exons with evidence of translation are more disordered than would be expected, we find contrasting results for tissue specific splicing events. The set of protein level tissue specific splice events actually have proportionally fewer disordered regions than non-tissue specific splice events.

There is some overlap between our data set and the exons used in these two analyses. For example, *BIN1*, illustrated in the Ellis et al study [32], is part of our ASE255 set. However, our set is highly enriched in exons that evolved during or prior to the vertebrate clade and more recently evolved splice events are significantly enriched in predicted disordered regions (S2 Fig). Recently evolved splice events have significantly more disordered regions than those that evolved more than 400 million years ago (Fisher exact test value < 0.00001). Although tissue specific alternative splicing is likely to affect protein-protein interactions, our study suggests that the role of disorder may not be as important as has been suggested.

Most protein level tissue enrichment at the protein level occurred in either muscle or nervous tissues. By way of contrast to other analyses [22,24], which found considerable evidence of tissue-specific splicing in testis at the protein level, we detected little evidence for tissue-specific splicing at the protein level in testis or in grouped reproductive tissues as a whole. Very few events were significantly enriched at the protein level in reproductive tissues and more than a third of the 71 events enriched at the transcript level were actually depleted at the protein level.

Nervous and cardiac tissues have been shown to have an important number of conserved tissue-specific splice events [33,34]. Our protein-level results are in agreement with an analysis of transcript level splicing signatures across multiple vertebrate species [2], which found that brain and heart/muscle tissues had strong conserved splicing signatures, while remaining tissues clustered by species rather than by tissue.

Functional analysis showed that protein level tissue-specific events were significantly enriched in genes annotated with functional terms related to the cytoskeleton. Genes with significant tissue-specific alternative splicing in muscle tissues (principally heart) were related to the composition and function of muscle and the Z-discs in the sarcomere, while genes with significant tissue-specific alternative splicing in nervous tissues were related to cytoskeletal connections and cell-cell contacts.

The importance of tissue-specific alternative splicing in two specialised tissues like brain and heart, the clear evidence of deep conservation, and the functional terms that are associated with the cytoskeleton and cellular differentiation paints a picture in which tissue-specific alternative splicing has been decisive in the development of nervous and muscle tissues. Our results are supported by previous data that document that tissue-specific splicing plays an important role in the development of brain and heart tissues [35–37].

In this study we have identified many functional alternative isoforms along with the tissues in which they are most expressed. The challenge is to determine exact functional roles for those isoforms where none is known. The gene *NEBL*, for example, has two main isoforms that differ in their N-terminals, the longer is called nebulette and the shorter LASP2. We find that nebulette is expressed exclusively in cardiac tissues, while LASP2 is found most often in nervous and urinary tissues and not in muscle tissues. Although the role of nebulette in binding Z-disc associated desmin filaments in cardiac tissues has been known for several years [38], LASP2 has only recently been shown to play a crucial role in post-synaptic development in the brain [39]. In order to further the investigation into the roles of these undoubtedly important alternative isoforms, we have listed many of the tissue specific alternative isoforms analysed in this study on the APPRIS web site [20].

## Material and methods

### Human reference genome

This study was based on the annotations in v27 of the GENCODE human reference gene set. The manual annotations in GENCODE v27 [3] are equivalent to Ensembl 90 and were produced in June 2017. The GENCODE v27 gene set had 19,881 protein coding genes.

### Proteomics analysis

We reanalysed the data from the Kim *et al* [21] proteomics experiments. The data comprised spectra from high-resolution Fourier-transform mass spectrometry experiments of 30 histologically normal human samples, including 17 adult tissues, 7 foetal tissues and 6 purified primary haematopoietic cells. In total there were 79 usable experiments, 18 covering fetal tissues and 61 covering adult tissues and haematopoietic cells. All tissues had at least two replicate

experiments, though the number of replicates varied. Adult heart had five replicates, for example.

Spectra from each experiment were downloaded from ProteomeXchange [40] and were searched against the GENCODE v27 human reference proteome, a decoy database [41] and a list of common contaminants, using the COMET search engine [42]. COMET allowed fixed post-translational modifications of methionine. The peptide spectrum matches (PSMs) from COMET were post-processed with Percolator [43]. We were more interested in reducing false positives than in increasing coverage, so we selected those PSMs that had a posterior error probability (PEP) lower than 0.001. PEP values of less than 0.001 in our analysis equated to PSM q-values of less than 0.0001. In addition, peptides were also limited to those that were fully tryptic, had no more than a single missed cleavage and had a length between 7 and 40 residues. Peptides that mapped to more than one gene were also discarded. With these rules in place we identified at least 2 PSM for 11,065 coding genes in the GENCODE v27 reference set.

Although we searched for tissue specificity using the 30 distinct tissues in the Kim *et al* analysis, much of the analysis was based on pooling the 30 tissues and hematopoietic cells into 10 groups of related tissues. This was done to amplify any signal. The proteomics tissue groups are detailed in **S3 Table**.

### Alternative splicing analysis

We analysed the tissue specificity of splice events rather than the tissue specificity of entire transcripts and splice isoforms because RNAseq reads and peptides are too short to cover more than short regions of sequence. Transcript reconstruction methods can be used to predict alternative transcript levels, but these methods are inaccurate [44] and there is no equivalent method for proteomics data.

In order to analyse splice events it is necessary to introduce the idea that splice events have two sides. Discriminating peptides and RNAseq reads will map to one side or the other of a splice event. For example, in the case of an indel one side of the event will be an insertion and the other a deletion, while there will be two different amino acid sequences at the protein level in the case of substitutions. We distinguished each side of the event as the main (the side of the event with most protein evidence) or alternative.

Analysis of the proteomics data allowed us to detect the presence or absence of peptides in distinct tissue-based experiments. Given the format of the experiments we were analysing (label-free experiments, replicates for all tissues) we chose to count the number of experiments in which splice event distinguishing peptides were detected. Each peptide was associated with a peptide-experiment detections (PEDs) count, which represented the number of experiments in which a peptide was detected. A peptide that was identified in every single experiment would therefore have 79 PEDs; peptides identified in a single experiment would be associated with just one PED. For the analysis we required that each side of a splicing event (main and alternative) was supported by a minimum of three PEDs (**Fig 8A**). This threshold was applied because it is not possible to detect the significance of tissue specificity for events supported by fewer than three PEDs.

### Protein level tissue specificity calculations

To carry out tissue-specific analysis at the protein level we annotated one side of each splice event as belonging to the main isoform, the side of the event with the most supporting PEDs, while the other side of the event was determined to belong to an alternative isoform. For each event, PEDs that supported each gene were separated into three types (**Fig 8B**), those that supported the side of the splice event with most evidence (that would give rise to the main

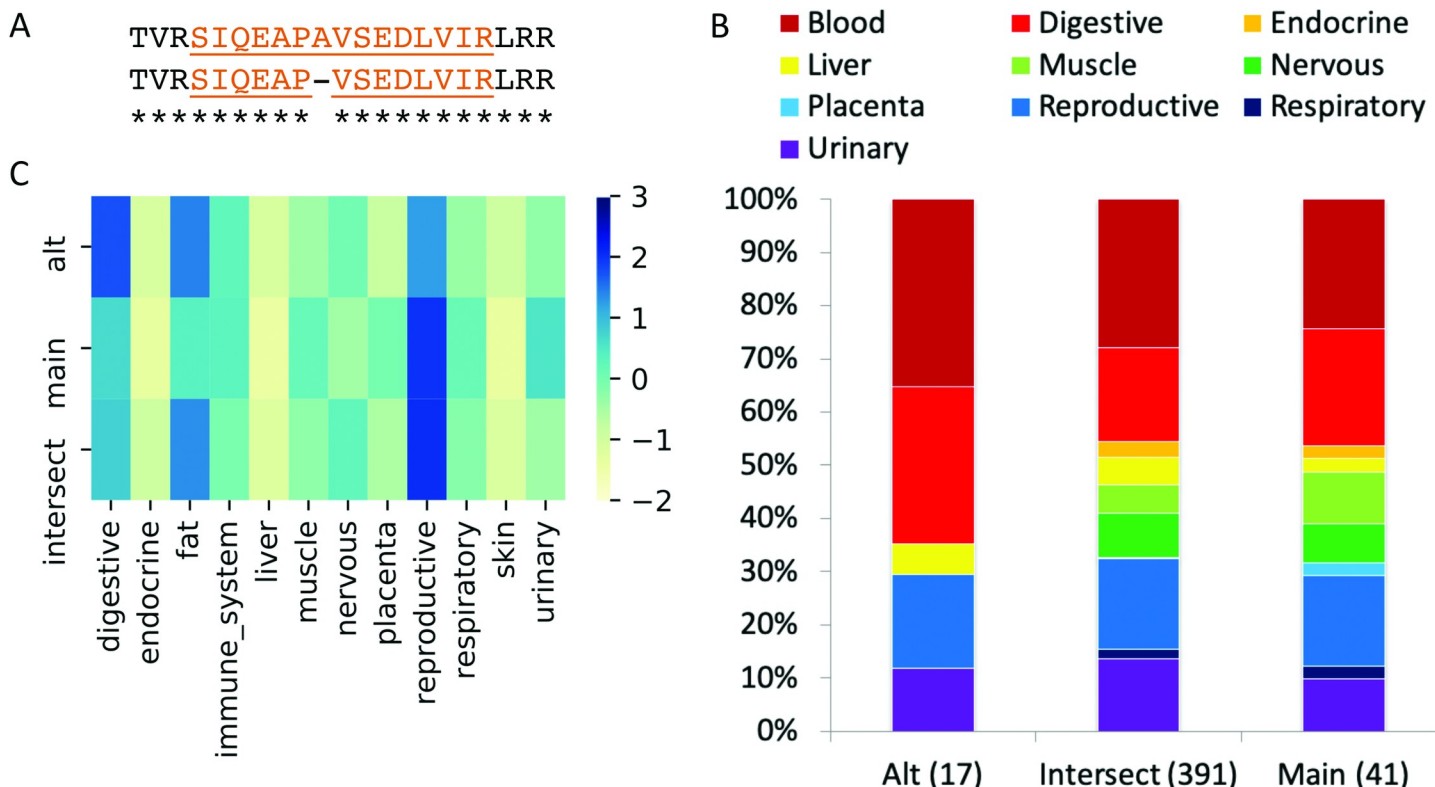

**Fig 8. Group-specific splicing in gene *TOR1AIP1*.** (A) We found peptide evidence for a single primate-derived splice event for *TOR1AIP1*. This NAGNAG splicing event resulted in the loss/gain of a single amino acid. The discriminating peptides we detected are highlighted. (B) The distribution of the PEDs for the discriminating peptides ("Alt" and "Main") and the remaining peptides that mapped to TOR1AIP1, but that did not distinguish one isoform from the other ("Intersect"). The number of PEDs in each set is shown in brackets. Fisher tests show that the distributions of PEDs between the Main and Alt peptides are not significantly different over any of the ten tissue groups. (C) The group-specific distribution of reads that support each side of the splice junction (main, alt) and those that support remaining common protein sequence (intersect) coloured by standard deviation from the mean; the darker the colour, the greater the positive standard deviation. There is more than one standard deviation between the reads for the digestive and reproductive groups, so the *TOR1AIP1* event is determined to be group specific at the transcript level for these tissue groups.

isoform), those that supported the other side of the splice event (the alternative isoform) and those that did not discriminate between the main isoform and the alternative isoform (the intersect).

We used the PEDs to calculate three sets of contingency tables for Fisher's exact tests, always comparing one tissue or group against the rest of tissues or groups. Fisher's exact tests were carried out for all tissues between main isoform and alternative isoform, main isoform and intersect, and alternative isoform and intersect.

### Transcript expression data

We downloaded data from the large-scale RNAseq analysis carried out by the Human Protein Atlas [31]. The RNAseq analysis was performed on 36 different tissues. It covers similar tissues to the Kim *et al.* analysis, though this analysis did not investigate fetal tissues or blood cells (S3 Table). We aligned the RNAseq data to GENCODE v27 using STAR 2.6 [45], forcing end-to-end read alignments to avoid unwanted alignments to repetitive regions. The maximum number of multiple alignments allowed was 50 and the rest of parameters were set by default.

We grouped tissues from the Human Protein Atlas analysis into 12 groups, where possible using just those tissues analysed in the proteomics experiments. Transcriptomics analysis

tissue groupings are shown in S3 Table. Tissues not interrogated in the proteomics experiments (such as skeletal muscle and duodenum) were left out of the groupings. There were three groups that did not appear in the proteomics analysis (skin, fat and immune system) and one proteomics analysis group that did not have an equivalent in the transcriptomics analysis (blood).

For the 255 splice events in the protein level alternative splicing set, we summed the reads into three groups in the same way that we did for the peptides, those reads that distinguished either the alternative or the main side of the splicing event and those that did not distinguish either side of the event. We calculated the mean number of reads across all the tissue groups for each gene and used the mean to calculate standard deviations for each set of reads that mapped to each peptide (Fig 1C). Events were counted as tissue group specific when the reads that mapped to one side of the splice event (equivalent to the main protein isoform or the alternative splice isoform) were at least one standard deviation higher than the other side of the splice event. Heat maps for the splice events in the ASE255 set are shown in S7 Fig.

## Alternative exon age

We calculated the age of the splice events in the ASE255 set manually by searching for supporting evidence in the UniProtKB database. We carried out BLAST [46] searches against vertebrate sequences with the residues that made up each side of the event and manually noted the presence or absence of the required sequence. If the event was shorter than 20 amino acid residues, we added flanking amino acids so that the search sequence was at least 20 amino acids long. We complemented BLAST searches with multiple alignments of vertebrate sequences.

To analyse the age of events in the genome as a whole, we calculated cross-species conservation scores for the alternative exons in the human reference set. Alternative exons were defined at the genome level using the APPRIS database [20]. APPRIS selects a representative protein isoform as the principal isoform for every coding gene. APPRIS determines principal isoforms based on protein structural and functional information and a score representing cross-species conservation and we have demonstrated that a single main isoform is the reality for the majority of coding genes and that APPRIS is the best predictor of this main isoform [7]. Alternative isoforms were all isoforms that were not tagged as principal. Alternative exons were those that did not overlap at all with exons that produced principal isoforms.

Ideally we would also calculate exon age manually, but this is not feasible at the genome level. Instead we calculated exon age from the cross-species conservation of the amino acid sequence corresponding to each exon. Cross-species conservation was calculated from BLAST searches against a protein database. We limited our analysis to alternative exons with a minimum of 42 bases to reduce the error rate.

Searches were carried out in two ways. Firstly, we searched for similarity to the translated exon itself, and secondly, we searched for similarity to the translation of the exon joined to the neighbouring exon (in the case 3' and 5' exon substitutions), or the exon plus both flanking exons (in the case of inserted or substituted exons). For searches with both sets of exons we recorded the species of those homologous sequences that had fewer than four residue insertions. The most distant homologue in each search was taken to represent the predicted age of the exon. The final exon age was the minimum of the predicted ages in the two analyses (the single exon and multiple exon calculations).

## Disorder predictions

We downloaded the IUPred2 disorder predictor [47] to make predictions for disorder for the splice isoforms in the ASE255 set. We calculated long disorder for all regions that differed

between the main and alternative isoforms. For indels we calculated disorder for the insertion, for substitutions we calculated disorder for both regions involved in the swap and took the region with the highest proportion of disorder as the representative score for that event. Events that were four or fewer amino acids in length were left out of the analysis. IUPred defines a disordered residue as having a score of 0.5. We defined a region as disordered if more than half of the amino acid residues scored more than 0.5.

### GO term calculations

We used DAVID [48] to calculate the significantly enriched GO terms within the genes we detect alternative splicing for, within those genes that had tissue-specific alternative splicing in nervous tissue in both proteomics and transcriptomics experiments, and within those genes that had tissue-specific alternative splicing in muscle tissue in both proteomics and transcriptomics experiments. As a background we used the 10,485 genes that we detected in the Kim *et al* experiments that had at least two distinct non-overlapping peptides. This was to remove in-built biases of the proteomics experiments and to limit to those genes for which it was minimally possible to detect two distinct splice isoforms.

## Supporting information

**S1 Fig. Significant tissue specific alternative splicing cases in proteomics tissues.** The count of the number of times we recorded tissue specific differences at the protein level in each of the 30 tissues.
(PDF)

**S2 Fig. Predicted order and disorder for alternative exons.** Mean order and disorder predicted by IUPred for various subsets. *Protein TS* are those events that are tissue specific at the protein level. *Transcript TS* are those events that are tissue specific at the transcript level. *Cassette TS* are skipped exon events that are tissue-specific at the protein level. *Protein Not* are those events that are not tissue specific at the protein level. *Transcript Not* are those events that are not tissue specific at the transcript level. *Cassette TS* are skipped exon events that are not tissue specific at the protein level. *Ancient* are those events that manual curation has shown to evolve more than 400 million years ago. *Recent* are all other events.
(PDF)

**S3 Fig. The relative ages of splice events in cytoskeleton-related genes.** The number of events with evidence in four different clades (vertebra to primates) separated into four groups by whether or not they were present in cytoskeleton-related genes ("Cytoskeleton" and "Other genes"), and whether or not the event was found to be significantly tissue specific at the protein level ("TS" or "Not"). There was a significantly higher proportion of vertebrate-derived events among the tissue specific events in cytoskeleton-related genes (Fisher's exact tests: 0.0093 vs Other genes TS, less than 0.00001 for the other two non-tissue specific groups).
(PDF)

**S4 Fig. The number of events that were tissue-specific in each of the 12 transcriptomics tissue groups.**
(PDF)

**S5 Fig. Correlation between supporting PEDs and supporting reads.** For each enriched/depleted event in the corresponding tissue the chart shows the percentage of reads support one side of the event that are detected in the corresponding tissue, plotted against the percentage of all PEDs for the same side of the event detected in proteomics experiments for that tissue.

Results are shown just for those events that are enriched/depleted in transcriptomics experiments in (A) digestive, (B) muscle, (C) nervous and (D) reproductive tissues.
(PDF)

**S6 Fig. Percentage of PEDs supporting the transcript level enrichment.** The figure shows the percentage of supporting PEDs for the four tested tissue groups (digestive, muscle, nervous and reproductive) from events are enriched (or depleted) in these groups in transcriptomics experiments. The percentage of supporting PEDs among all PEDs detected are shown for the sides of the events that are enriched in transcriptomics experiments (dark red) and for the sides of the events depleted in transcriptomics experiments (light blue). The percentage of PEDs are shown over all events enriched in transcriptomics experiments (*All*), over the subsets of events enriched in transcriptomics experiments that evolved after the split from fish (*Tetrapoda*) and over those that evolved after the split from monotremes (*Theria*). The number of events enriched in transcriptomics experiments and in each subset is shown in the x-axis. Asterisks above the bars show where the number of PEDs supporting the enriched side of the events were significantly different from the number of PEDs on the depleted sides of the events as would be expected if the events were group specific as a whole.
(PDF)

**S7 Fig. Heatmaps with the standard deviation for the AS events.** The darker the colour, the greater the standard deviation. We calculated the mean number of reads across all the tissue groups for each gene (*intersect*) and used the mean to calculate standard deviations for each set of reads that mapped to each event. Events were counted as tissue group specific when the reads that mapped to the *main* or alternative (*alt*) side of the splice event were at least one standard deviation higher than the other side of the splice event.
(PDF)

**S1 Table. The ASE255 set.**
(XLSX)

**S2 Table. GO terms for tissue specific events.**
(XLSX)

**S3 Table. List of groups of tissues in proteomics and RNA-seq experiments.** Human Body Map tissue proteomics experiments collected in tissue groups (tab 1) and Human Protein Atlas tissue transcriptomics experiments collected in tissue groups (tab 2).
(XLSX)

## Acknowledgments

The authors would like to thank Federico Abascal for his invaluable input on this paper.

## Author Contributions

**Conceptualization:** Michael L. Tress.

**Data curation:** Jose Manuel Rodriguez, Fernando Pozo, Tomas di Domenico.

**Formal analysis:** Jose Manuel Rodriguez, Michael L. Tress.

**Funding acquisition:** Jesus Vazquez, Michael L. Tress.

**Investigation:** Jose Manuel Rodriguez, Michael L. Tress.

**Methodology:** Jose Manuel Rodriguez, Jesus Vazquez, Michael L. Tress.

**Project administration:** Michael L. Tress.

**Resources:** Fernando Pozo, Tomas di Domenico.

**Software:** Jose Manuel Rodriguez.

**Supervision:** Jesus Vazquez, Michael L. Tress.

**Validation:** Jose Manuel Rodriguez, Michael L. Tress.

**Visualization:** Jose Manuel Rodriguez, Michael L. Tress.

**Writing – original draft:** Jose Manuel Rodriguez, Michael L. Tress.

**Writing – review & editing:** Jose Manuel Rodriguez, Fernando Pozo, Jesus Vazquez, Michael L. Tress.

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
