## [Decision Letter · Decision Letter 0]

20 Jun 2020

Dear Dr Tress,

Thank you very much for submitting your manuscript "An analysis of tissue-specific alternative splicing at the protein level" for consideration at PLOS Computational Biology.

As with all papers reviewed by the journal, your manuscript was reviewed by members of the editorial board and by several independent reviewers. In light of the reviews (below this email), we would like to invite the resubmission of a significantly-revised version that takes into account the reviewers' comments.

In particular, it is important that you clarify the datasets and the backgrounds used. It is also important to discuss fully any confounding issues and to cite all appropriate references in your introduction and discussion.

We cannot make any decision about publication until we have seen the revised manuscript and your response to the reviewers' comments. Your revised manuscript is also likely to be sent to reviewers for further evaluation.

Sincerely,

Christine A. Orengo

Associate Editor

PLOS Computational Biology

Arne Elofsson

Deputy Editor

PLOS Computational Biology

Reviewer's Responses to Questions

**Comments to the Authors:**

Reviewer #1: Rodriguez and colleagues study tissue-specific alternative splicing (AS) at the proteomic level. By re-analyzing the large-scale proteomics dataset from Kim et al, the authors perform one of the first (if not the first) studies evaluating tissue regulation of AS directly at the proteomics level. While the data is a bit limited (255 high-confidence events), it is still of interest and some relatively clear patterns emerged. I would thus support publication of this study if a few issues were addressed:

1) As it could be expected given the history of this topic (the proteomic impact of AS), the Introduction is a bit unbalanced. Many of the cited papers are from the senior author, while certain papers, especially those from the Blencowe lab, are missing (e.g. the discussion on Trends in Biochemical Sciences, the evolutionary studies, various papers on splicing regulation and function, etc.). Given that this is a Results (and not an Opinion) article, it will be important that the authors make an extra effort to be as objective and balanced as possible.

2.1) Overall, it is complex to figure out which are the 255 AS events the authors are studying. A text table with the AS events, with their protein and genomic coordinates, and the peptide counts in each tissue sample should be provided.

2.2) Also, it would be helpful if the authors also report the events that did not fully met their threshold criteria (at least 3 PEDs for each variant), since AS events with 2 PEDs could also be very interesting. For these cases, I wonder: if a given tissue (e.g. Adult Frontal cortex) has only two replicates, would all tissue-specific AS events from that tissue be systematically excluded (i.e. those whose alternative isoforms are only detected in that tissue)? This could be artificially reducing the fraction of tissue-specific AS events detected. Therefore, the authors should consider analyzing also those AS events with PEDs in all the replicates of a given tissue (even if these have only 2 PEDs in total). This will expand the small number of studied events (currently 255), which is a weakness of the study.

3) The section "Tissue specificity by event type" and Figure 2 are weak. The categories considered by the authors might be the most functionally relevant from the proteomics perspective, but are difficult to relate to the standard AS event types (cassette exons, mutually exclusive exons, alternative splice site donors/acceptors) and to other forms of transcriptomic variation (e.g. alternative first or last exons by alternative promoters and polyadenylation sites). Figure 2 should include the data split by these event types as well.

4) The study of the age of the events should also be improved. First, it is unclear which species were searched to define ancient events. Most importantly, the comparison with their genome background does not seem fair. If I understood correctly, they defined as "alternative events" those sequences that do not overlap the principal isoform defined in APPRIS. While this is a good idea, it misses an important group of AS events: those sequences that are generally included, but that are skipped in a (smaller) fraction of transcripts. These AS events will be more ancient according to their definition. The easiest way to assess the impact of this omission is to see for how many of the 255 events the alternative sequence is in the principal isoform. Only those that do not overlap the principal isoform can be fairly compared with their current genome background. Then, ideally, the authors should also come with another background set: sequences in the principal APPRIS isoform that are missing in any of the non-principal isoforms. And determine their evolutionary age, for comparison. Finally, as a minor point, the exclusion of exons smaller than 42 nucleotides is also expected to introduce biases, since small exons have been shown to be particularly conserved and tissue-specific.

5) Their results suggest that the digestive tissue RNA-seq they analyzed is problematic, and this may be affecting the transcriptomic comparison. For example, the conclusion that "the tissue group with the most evidence for group-specific expression is digestive tissues" is at odds with every previously published study. Hence, I suggest that the authors use other digestive tissue RNA-seq, even if they are from a different source, to perform the transcriptomic vs proteomic comparison.

6) I found that the transcriptomic vs proteomic comparison is a bit convoluted and difficult to follow. Could this be simplified in the main text? In the end, their point comes across, but I wonder whether all the details to get the PGE99 and TGE355 sets are really needed in the main text. Also, as mentioned above, the result comparing proteomics to transcriptomics is clear, but the opposite is not, likely due to the noisy nature of some of the RNA-seq samples. As suggested above, perhaps by using better transcriptomic samples it will be possible to spare the reader the details of Figure 6, which really have little to do with the rest of the study and does not add much. Finally, I wonder: do measures by RNA-seq and mass-spec correlate? E.g. the z-scores across tissues of the ratio of isoforms (or "sides").

7) A formal point: I found the concept "side of the splicing event" is a bit mis-leading. What does "side" refer to here? Is there a better way to refer to this? What about simply "variant"? The first time I read I thought this was referring to the two sides of an exon (upstream and downstream exon junctions).

Reviewer #2: I found this work on the whole interesting and well put together. The functional role of alternative splicing is controversial and I think this work helps to further identify a subset of minor isoform's that everyone can agree have promising functional signatures. Thanks to the authors for providing this interesting analysis that I enjoyed reading.

Requested Revisions

You need to detail and then justify the background sets you used for the DAVID analysis. I think in this case its just a matter of being more explicit. It could be that you used the whole genome for the background. If this has been done, then I don't think this would be appropriate background. So some clarification on the backgrounds is needed.

With regards to the accessibility of the paper, its good to keep in mind that many people reading the paper will not be quite as familiar with

the definitions. In Figure2 B the different classes could have expanded explanations as to how they are precisely defined to make the results more accessible (in a supplementary table). For example can C-terminal N-terminal potentially come from variable promoter usage. That's not a problem if they do, but further detailed qualifications of what types of events could belong to each group would be helpful in a table.

I suspect there could be some confounding factors that aren't fully addressed. For example, short indels are less tissue specific but also presumably very difficult to detect through homology searches (so will rarely be identified as conserved even if they are) . This would then make tissue specific events, enriched in homologous exons, appear to be more ancient because the homologous exons are so much easier to detect using sequence searches. I may have interpreted this wrong but I would be interested to hear from the authors about this. If there are genuine caveats here perhaps some extra text on this could be warranted in the caveats section.

Suggested

Missing full stop after this sentence "but most were generated by NAGNAG splicing [38]" also perhaps have a sentence defining NAG NAG splicing e.g. but most were generated by NAGNAG splicing [38] a form of splicing where....

Figure4 B the color charts are difficult to follow because they are all shades of maroon. You may want to think about making the color bars more distinguishable for different tissues.

I didn't quite follow the logic of this sentence "If conserved post-translational

buffering systems exist, this would remove the need for transcript level constraints." I think expanding the sentence to spell out the meaning would be helpful.

Figure 1 seems quite stranded in the methods section and is only referred to as figure 1C from the text. Usually its better if all aspects of a figure are referred to from the text. Otherwise it suggests the figure needs reworking. I think an example here is good, but it needs work to integrate it into the text more.

In the Author summary I felt this sentence needed improving:

"and the three quarters of primate-derived alternative exons may not behave the same way."

In the following section it may be important to bring in the idea and contribution of other processes such as alternative promoter usage.

"This is reflected in the human reference set; at present human coding genes are

annotated with an average of four distinct gene products [3]. Recent studies suggest that human coding

genes generate on average more than ten alternative transcripts [4, 5]"

Replace Gtex with GTEx

What looks like a minor omission in the analysis is how this work closely relates to

https://www.ncbi.nlm.nih.gov/pmc/articles/PMC3437557/

Not that you would have to do any analysis, but you could have something in the conclusions, since one question for me that's left open is what are the alterations in protein structure for the tissue specific events relative to the Babu paper.

General comments

I found the text on the whole very readable. However, the section "Event age and tissue groups" was difficult to wade through. I would suggest distilling this section down a bit, because as it stands some readers may struggle and not pick out the key results and their implications.

I liked the way that you put some emphasis on the need to functionally characterize a subset of splicing events that look functional. With the advent of CRISPR technologies experiments the ability to investigate these functions is becoming far easier. Given your groups expertise I wonder if the paper from this data could be combined with other data in APPRIS to provide a webpage formatted as a data table entitled "subset of minor isoforms recommended for functional characterisation" . Tissue specific proteomic detected isoforms could be one way of adding entries to this table. This is just a suggestion and would require some work and isn't necessary for publication. But I think would add to the paper if the current submission is not successful.

Reviewer #3: The paper addresses one of most central questions of the eukaryotic biology – what is the role of alternative splicing? Many opinions concerning this topic have been expressed, ranging from ones that alternative splicing is what enabled the evolution of multicellular organism, to ones that it is largely noise. However, there are few rigorous analyses providing evidence one way or another, thus every little step in this field is important. The authors make such a step by reanalysing proteomics data from Pandey lab published in a somewhat controversial paper in 2014, in combination of transcriptomics data from the Human Proteome Atlas.

The bioinformatics analysis is generally done thoroughly, and the manuscript written clearly, however I found a couple of concerns or analysis omissions, which I think can be addressed.

1) What about the change in the overall expression levels of a gene between conditions between which the alternative splice event happen? If the overall gene expression level in one condition is different from that in the other by an order of a magnitude, then the interpretation of the alternative splicing event would be rather different from the cases where in both conditions the gene overall expression level (i.e., summing up all splice forms) is of the same magnitude. The first case may be easily noise. How often each type of events happens? Which are the events of the second category? Making this assessment on proteomics level may be tricky, however at least on RNA level this should be done and added to the analysis.

2) In the Results subsection “Comparison to tissue-specific splicing at the transcript level”, I didn’t find a clear answer to the (in my mind) most important question: what proportion of the tissue specific alternative splicing events observed at the proteomics level have a clear support on RNA level?

3) The authors rightly say that the some of the conclusion of the paper are confounded by the proteomics and RNA datasets are not being paired – the samples analysed are different. Am I wrong to think that in the last few years Human Proteome Atlas have done their own proteomics assays and the proteomics data are available? May be for some reason these data are not appropriate for the analysis that the authors do, however it must be possible to find an additional proteomics dataset for at least two of the tissues. The paper would be much stronger if some of the analysis could be replicated on a second dataset.

4) I would not draw any conclusions regarding the testis tissue: many more genes are expressed in testis on high levels than in other tissues, which will confound any comparisons.

5) The initial part of the Conclusions largely repeats what was already said earlier in the paper. What really are the main conclusions? Possibly it is difficult to draw strong definite conclusions from any individual analysis in this field, but may be the authors should use a part of the Discussion section for a more general discussion. I found their statement that “Instead, our results suggest that much tissue-specific alternative splicing has been involved in the evolution of vertebrate organs, principally heart and brain” important, and perhaps the main specific conclusion. Maybe the authors want to elaborate on this.

6) It has to be noted that the author found only a small number of events, however this fact is not reflected in Abstract. There is even smaller number of events found affecting the “interior” of the gene (9?). Even if these small numbers are in partly caused by the limitations of the available data, it is important to note that “more than the third of events detected” actually correspond to a rather small number overall number. When I was reading the Abstract for the first time, I reached a very different conclusion than after reading the whole manuscript. The conclusion really is that among the small number of alternative splicing events that the authors were able to detect in the proteomics data, a third are tissue specific (or perhaps less, if we account for differences in gene expression, see item 1) above). This is not a criticism, mostly likely the number of functionally relevant alternative splicing events really is small.

I minor comment. I was not able to understand Figure 1B. What do the colours use

**Have all data underlying the figures and results presented in the manuscript been provided?**

Reviewer #1: No: I could not find the peptide counts for each event, for instance.

Reviewer #2: No: Its probably just me but I couldn't find the links to:

S1 Appendix. Supporting figures and tables.

Supplementary tables 1 and 2, and supplementary figures 1-6.

Reviewer #3: Yes

PLOS authors have the option to publish the peer review history of their article (what does this mean?). If published, this will include your full peer review and any attached files.

Reviewer #1: No

Reviewer #2: No

Reviewer #3: No
---

## [Decision Letter · Decision Letter 1]

25 Aug 2020

Dear Dr Tress,

We are pleased to inform you that your manuscript 'An analysis of tissue-specific alternative splicing at the protein level' has been provisionally accepted for publication in PLOS Computational Biology.

Best regards,

Christine A. Orengo

Associate Editor

PLOS Computational Biology

Arne Elofsson

Deputy Editor

PLOS Computational Biology

Reviewer's Responses to Questions

**Comments to the Authors:**

Reviewer #1: The authors have satisfactorily addressed most of my concerns.

Reviewer #2: Thanks for the interesting responses to the questions (Especially the addition of the website that makes the results available for further functional characterization.)

Reviewer #3: I have reviewed the revision and all my concerns have been addressed.

**Have all data underlying the figures and results presented in the manuscript been provided?**

Reviewer #1: None

Reviewer #2: Yes

Reviewer #3: Yes

PLOS authors have the option to publish the peer review history of their article (what does this mean?). If published, this will include your full peer review and any attached files.

Reviewer #1: No

Reviewer #2: No

Reviewer #3: **Yes: **Alvis Brazma

---

## [Editor Report · Acceptance letter]

28 Sep 2020

PCOMPBIOL-D-20-00204R1 

An analysis of tissue-specific alternative splicing at the protein level

Dear Dr Tress,

I am pleased to inform you that your manuscript has been formally accepted for publication in PLOS Computational Biology. Your manuscript is now with our production department and you will be notified of the publication date in due course.

With kind regards,

Sarah Hammond
